# Mitral Atresia with Normal Aortic Root

**DOI:** 10.3390/children9081148

**Published:** 2022-07-30

**Authors:** P. Syamasundar Rao

**Affiliations:** Children’s Heart Institute, Children’s Memorial Hermann Hospital, McGovern Medical School, University of Texas-Houston, Houston, TX 77030, USA; p.syamasundar.rao@uth.tmc.edu; Tel.: +1-713-500-5738; Fax: +1-713-500-5751

**Keywords:** mitral atresia, patent foramen ovale, single ventricle, balloon atrial septostomy, blalocktaussig shunt, pulmonary artery banding, inter-stage mortality, bidirectional Glenn, Fontan operation

## Abstract

Mitral atresia with normal aortic root is a rare complex congenital heart defect (CHD) and constitute less than 1% of all CHDs. In this anomaly, the mitral valve is atretic, a patent foramen ovale provides egress of the left atrial blood, either a single ventricle or two ventricles with left ventricular hypoplasia are present, and the aortic valve/root are normal by definition. Clinical, roentgenographic and electrocardiographic features are non-distinctive, but echo-Doppler studies are useful in defining the anatomic and pathophysiologic components of this anomaly with rare need for other imaging studies. Treatment consists of addressing the pathophysiology resulting from defect and associated cardiac anomalies at the time of initial presentation, usually in the early infancy. These children eventually require staged total cavo-pulmonary connection (Fontan) in three stages. Discussion of each of these stages were presented. Complications are observed in-between the stages of Fontan surgery and following completion of Fontan procedure. Attempts to monitor for early detection of these complications and promptly addressing the complications are recommended.

## 1. Introduction

Mitral atresia may be seen in subjects with both aortic atresia or a normal aortic root; the first lesion is commonly designated as hypoplastic left heart syndrome (HLHS) which was reviewed elsewhere [1,2,3]. Mitral atresia with normal aortic root will be the subject of this review. Whereas this differentiation is arbitrary, it is thought to be appropriate [4] since there are significant variances of how we treat these two congenital heart defects (CHDs). The HLHS babies almost always need the Norwood procedure as neonates [5,6], while babies with mitral atresia with normal aortic root require different forms of therapy at different ages on the basis of associated heart defects [4,7].

Mitral atresia with normal aortic root is an uncommon, complicated CHD and is typically associated with multiple other heart anomalies. The occluded mitral valve may be in the form of an imperforate valve membrane or an absent atrioventricular (AV) connection [8]. In a study of 30 mitral atresia autopsy heart specimens, six specimens demonstrated an imperforate left AV valve while the remaining 24 showed an absent left AV connection [9]. Nevertheless, such a differentiation is not clinically important in that both types require similar treatment algorithms [4,7]. In children with ventricular inversion, the atretic AV valve is a morphologic tricuspid valve, as observed in hearts with tricuspid atresia, Type III, subtypes 1 and 5 [10]; nonetheless, the physiology in these patients is akin to that of mitral atresia. Therefore, using the term “left AV valve atresia” in such situations is acceptable.

## 2. Pathologic Anatomy

The left atrium (LA) is small in size and may be hypoplastic. The right atrium (RA) is dilated and hypertrophied. Either a patent foramen ovale (PFO) or an atrial septal defect (ASD) is usually present; a PFO in 2/3rds of the subjects and an ASD in the remaining 1/3rd [11]. Restrictive atrial defect and, rarely, an intact atrial septum may be seen; in such patients the levoatriocardinal vein [12] is likely to help decompress the LA. Most commonly, a single ventricle is present; however, two ventricles are seen in a few patients. In cases with two ventricles, the left ventricle (LV) is hypoplastic and connects with the right ventricle (RV) through a small ventricular septal defect (VSD). The RV is uniformly dilated and hypertrophied. The great arteries are frequently transposed [11,13,14]. Some infants may have double-outlet right ventricle. The pulmonary valve is normal and not stenotic in the majority of patients; though valvar and/or subvalvar stenosis or atresia may be seen in 25% to 30% of cases. Both the aortic valve and aortic root are normal in size by definition. Aortic coarctation or aortic arch interruption is present in approximately 30% of patients [15]; it has been observed that such aortic arch obstructions are found only in patients who have a normal pulmonary valve. Nearly 80% of patients have patent ductus arteriosus (PDA) [11].

## 3. Pathophysiology

Pulmonary venous blood from the LA goes through the ASD/PFO into the RA and mixes with the systemic venous blood. This produces systemic arterial desaturation in all children with mitral atresia. In patients with restrictive PFO, development of pulmonary venous congestion and edema may be seen. There appears to be a tendency for development of interatrial obstruction as these babies age. From the RA, the blood is conveyed via the tricuspid valve into the RV or single ventricle, as the case may be. This mixture of pulmonary and systemic venous returns is then distributed into the aorta and pulmonary artery dependent on their respective vascular resistances. The magnitude of arterial desaturation is inversely related to the degree of pulmonary stenosis (PS). In neonates with pulmonary atresia, the pulmonary blood flow is supplied through the PDA or rarely via the aortopulmonary collateral blood vessels.

## 4. Clinical Presentation

The types of clinical manifestations are largely related to the combined effect of the basic defect and associated heart abnormalities. The majority of patients manifest during the neonatal period, typically during the first week of life either with cyanosis or difficulty in breathing. Or, they may have been detected by fetal ultrasound during maternal pregnancy or by pulse oximetry screening after the baby is born.

### 4.1. Inter-Atrial Obstruction

Infants with inter-atrial obstruction exhibit signs of pulmonary venous congestion, namely, tachypnea, retractions, and cyanosis. In an occasional case with an intact atrial septum, the presentation is immediately following birth.

### 4.2. Severe Pulmonary Stenosis or Pulmonary Atresia

Infants who have severe pulmonary stenosis or atresia have diminished pulmonary blood flow (PBF) and their presentation is with cyanosis and dyspnea as the PDA begins to constrict. When the PBF is markedly diminished severe metabolic acidosis occurs.

### 4.3. No Pulmonary Stenosis

Infants without PS have elevated PBF and as the pulmonary vascular resistance (PVR) falls with aging (usually in a few weeks to months), volume overload occurs and congestive heart failure (CHF) develops. Tachypnea, tachycardia and cardiomegaly are the usual findings in these babies.

## 5. Physical Examination

On physical examination, the precordial impulses either show a right ventricular heave (in patients with PS or pulmonary atresia) or a hyperdynamic RV impulse (in subjects without PS). The second heart sound is single in patients with severe PS or pulmonary atresia while it may be split in babies without PS. Infants with PS exhibit long ejection systolic murmurs along the left upper sternal border. Such murmurs are absent in babies with pulmonary atresia. Children without PS may have non-specific ejection systolic murmurs of short duration along the left sternal border. Signs of CHF are perceived in the majority of babies without PS. In subjects with severe restriction of PFO, signs of pulmonary edema may be observed instead. Femoral pulses may be decreased in babies with coarctation or interruption of the aorta and as stated above, such aortic obstructions are seen in only infants who do not have PS.

## 6. Chest X-ray

There are no characteristic findings indicative of the defect on chest X-ray. However, chest roentgenographic features reflect the pathophysiologic status of PBF resulting from the combined effect of the defect itself and associated abnormalities. Mild cardiomegaly with decreased pulmonary vascular markings is seen in babies with severe PS or pulmonary atresia. Moderate to severe cardiac enlargement along with increased pulmonary vascular markings is seen in infants without PS. In babies with obstruction at PFO, pulmonary edema is seen on chest X-ray. The chest films are thus useful in assessing the status of PBF.

## 7. Electrocardiogram (ECG)

ECG demonstrates tall, peaked P waves in lead II and right chest leads suggestive of RA enlargement. Right axis deviation and RV hypertrophy are seen in most patients. On occasion, left axis deviation is seen (12% of the babies). Presence of Q waves in the right chest leads with a qR pattern was observed in most patients in one study [15]. The described ECG abnormalities are useful, but are not diagnostic for mitral atresia.

## 8. Echocardiogram

In contradistinction to findings on chest X-rays and ECGs, echo-Doppler studies are very valuable in making a diagnosis and in supplying suitable information for planning an appropriate treatment algorithm. The atretic left AV valve (Figure 1 and Figure 2) can easily be recognized irrespective of its morphology, namely, an absent connection (Figure 1A) or an imperforate membrane (Figure 1B and Figure 2). However, as mentioned above, such a distinction is not important in making the management decisions.

It is also imperative to identify if the defect is a part of complete AV septal defect; if such is the case, it is unlikely to have left atrial obstruction because AV septal defects typically have large ostium primum ASDs whereas mitral atresia cases have hypoplastic left atrium and potential to have obstructed inter-atrial defect.

The size of the ASD/PFO is appraised by imaging the diameter of the atrial defect by two-dimensional imaging (Figure 1B and Figure 3A) and by color Doppler imaging (Figure 3B). Turbulent flow (Figure 3B) and high mean Doppler gradient by pulsed and/or continuous wave Doppler are suggestive of interatrial obstruction.

After defining the size of the LA, atretic mitral valve and the atrial septum, one should evaluate if there are two ventricles or if there is only one ventricle, i.e., single ventricle. If there are two ventricles, the nature of ventricular looping (d-loop or l-loop), the size of the VSD, and where the aorta and pulmonary artery arise should be appraised. Then, the great artery relationship should be determined. In both the above groups, the presence of stenosis or atresia or wide-open (non-stenosed) pulmonary valve should be defined. In patients with PS, the magnitude of narrowing is assessed by continuous wave Doppler recording. In patients with pulmonary atresia, patency of the ductus arteriosus should be scrutinized. Suprasternal notch views should be performed and Doppler interrogation should be undertaken to identify aortic coarctation or interrupted aortic arch.

## 9. Cardiac Catheterization with Angiography, Magnetic Resonance Imaging (MRI) and Computed Tomography (CT)

Catheterization and angiography are not necessary since the diagnosis and pathophysiology of the lesion complex can be defined by echo studies. Same thing holds true for MRI and CT studies. However, catheterization is an integral part of relief of obstruction in patients with restrictive ASD/PFO and will be reviewed in the section on “Inter-atrial Obstruction”, under “Therapy”.

## 10. Therapy

### 10.1. Therapy at Initial Presentation

The management at the time of presentation, usually in the neonatal period, is similar to that used for other cyanotic CHD patients, as described elsewhere [16,17,18] and includes, circumventing development of hypothermia; maintaining neutral thermal environment; detecting and apt management of hypoglycemia and hypocalcemia; the observation of acid-base status; addressing metabolic acidosis with sodium bicarbonate (NaHCO_3_); and treating respiratory acidosis with suction, intubation, and assisted ventilation as deemed appropriate. If ductal dependent situation is suspected, the intravenous prostaglandin E_1_ (PGE_1_) should be started while waiting for confirmation of the diagnosis.

As stated in the preceding section, the clinical features are largely dependent upon the status of the PBF (presence of pulmonary outflow tract obstruction and the patency of the ductus), PVR, and the magnitude of restriction at the ASD/PFO level. Consequently, the management should address these problems. Following stabilization of the infant, the treatment is designed to address the pathophysiologic abnormality created by the defect complex, such as inter-atrial obstruction, reduced PBF, elevated PBF with or without CHF, or increased PBF combined with inter-atrial obstruction.

#### 10.1.1. Inter-Atrial Obstruction

In patients with restrictive ASD/PFO, producing pulmonary venous congestion, balloon atrial septostomy [19,20,21,22], initially described by Rashkind [19] should be performed. Balloon atrial septostomy may not be feasible in all babies since the LA is small and hypoplastic in the majority of these children. Static balloon dilatation [23,24] of the restrictive atrial defect with a balloon angioplasty catheter designed for balloon valvuloplasty (Figure 4) is an excellent alternative in such patients. The author prefers static balloon dilatation procedure because it not only relieves pulmonary venous congestion, but also retains some degree of restriction to transeptal flow which will not allow rapid fall in the PVR. Consequently, static balloon dilatation is a better choice in most patients.

Some infants, though rare, may have a very tight PFO (Figure 5) or even an intact atrial septum (Figure 6). In the intact atrial septum patients, puncturing the atrial septum by the Ross/Brockenbrough technique [25,26] or by radiofrequency perforation [27], followed by static balloon angioplasty of the atrial septum [23,24] may become necessary. Alternatively, particularly in cases with intact atrial septum or an extremely tight PFO (Figure 5), a stent may be implanted (Figure 7 and Figure 8) across the atrial septum [20,21,28]. If the transcatheter techniques are not possible or do not achieve the desired effect, atrial septectomy by surgery may be needed.

#### 10.1.2. Decreased Pulmonary Blood Flow

In babies with decreased PBF either due to pulmonary atresia or very severe PS, infusion of PGE_1_ intravenously should be started promptly; we usually begin with a dose of 0.05 to 0.1 mcg/kg/min and once the O_2_ saturation improves, the dosage is gradually reduced to 0.015 to 0.02 mcg/kg/min. Then, a more permanent way to supply pulmonary flow, usually via a modified Blalock-Taussig (BT) shunt [29,30] (Figure 9A and Figure 10) is performed. Alternative approaches are placement of stents within the ductus [31,32,33] and balloon pulmonary valvuloplasty in patients with predominant obstruction at the level of pulmonary valve [34,35,36]. In children with less severe degrees of PS with mild decrease in O_2_ saturations (in the low 80 s), no intervention is indicated.

#### 10.1.3. Increased Pulmonary Blood Flow with Heart Failure

In infants without PS, i.e., a normal pulmonary valve, as the PVR decreases with aging, the PBF increases and when this becomes excessive, symptoms of CHF appear. These babies are managed with anti-congestive measures [39]. Following control of the CHF, banding of the pulmonary artery [37] is performed (Figure 9B and Figure 11). Pulmonary artery banding helps achieve a better control of the CHF, reduces pressures in the pulmonary artery and prevents the onset of pulmonary vascular obstructive disease (PVOD).

#### 10.1.4. Increased Pulmonary Blood Flow without Heart Failure

In some infants without PS, there is gradual decline in the PVR and therefore, the rate of increase of the PBF is slow. Consequently, these babies tolerate this increased PBF and do not go into CHF. These infants should also have pulmonary artery banding (Figure 9 and Figure 11) [37] on an elective basis between four to six weeks of age or as and when they are identified. Again, the reason behind such a recommendation for pulmonary artery banding is to reduce the pulmonary artery pressures so that they later become suitable candidates for bidirectional Glenn and Fontan procedures and to prevent the development of PVOD.

#### 10.1.5. Increased Pulmonary Blood Flow along with Inter-Atrial Obstruction

This combination does occur commonly in this defect complex [22]. A fast and predictable decline in PVR happens in infants who have had relief of a restrictive ASD/PFO, whether it was done by balloon atrial septostomy or by surgical atrial septectomy [22]. Therefore, pulmonary artery banding [37] should be undertaken without any hesitancy at the time of relieving the atrial septal restriction so as to reduce the probability of development of CHF, decrease the pulmonary artery pressure, and prevent PVOD; this would also facilitate future bidirectional Glenn and Fontan procedures that many of these patients require.

### 10.2. Follow-Up after Initial Palliation

Because of the complexity of this heart defect, it is imperative to periodically re-examine these infants. The atrial septal communication that is adequate early in the neonatal period may develop restriction with time. The atrial defect may also become restrictive even after septostomy. Consequently, intermittent clinical and echocardiographic assessment of the atrial defect is a must. Eventually, unobstructed left to right shunt across the atrial septum (Figure 12) should be established. Infants who have had pulmonary artery banding should also be assessed with Doppler studies to assure that there is adequate constriction of the pulmonary artery (Figure 11) and, if needed, cardiac catheterization studies may have to be undertaken to measure the pressures in the pulmonary artery. Lastly, infants who have had a BT shunt should be followed to ensure that the shunt is patent (Figure 9 and Figure 10).

### 10.3. Additional Surgery

Definitive intracardiac repair of mitral atresia with normal aortic root is not feasible because there is either a single ventricle is present or when two ventricles are present, the left ventricle is hypoplastic. Consequently, most of these patients are addressed with single-ventricle palliation strategy by Fontan procedure [4]. The concept of using the single functioning ventricle to support the systemic circulation and to connect the systemic veins to the pulmonary artery was originally proposed by Fontan, Kruetzer and their associates [40,41] in the early 1970 s for the management of children with tricuspid atresia. The procedure was subsequently adopted to treat other heart defects with one functioning ventricle, including mitral atresia with normal aortic root. A number of modifications of this concept were introduced, the techniques have changed and the age at which the Fontan is performed have evolved during last few decades, as summarized elsewhere [42,43,44,45,46,47]. Currently, the Fontan procedure is performed by total cavo-pulmonary connection (TCPC), introduced by de Leval and his colleagues [48]. The Fontan procedure can’t be performed in the neonates and young infants because of high pulmonary artery pressures and increased PVR. At the present time, the Fontan procedure is performed as a staged TCPC with an extra-cardiac conduit and a fenestration with succeeding closure of fenestration by transcatheter methodology [42,43,44,45,46,47]. The Fontan procedure is performed in three stages: Stages I, II, IIIA and IIIB. Stage I is at the time of initial presentation, usually in the neonatal period, as described in the preceding sections (Figure 4, Figure 5, Figure 6, Figure 7, Figure 8, Figure 9, Figure 10 and Figure 11). Stage II is a bidirectional Glenn procedure performed around the age of six months during which the superior vena cava is connected the right pulmonary artery end-to-side (Figure 13A and Figure 14A). Stage IIIA Fontan consists of re-directing the inferior vena caval flow into the pulmonary artery, usually through a non-valved conduit [49,50], normally one year after the bidirectional Glenn operation (Figure 13B–E and Figure 14B). A fenestration is placed between the conduit and atrial mass by most surgeons during Stage IIIA [51,52] (Figure 13F and Figure 14B). If there is residual atrial tissue, it should be removed either during Stage II or Stage IIIA to ensure unobstructed blood flow from the pulmonary veins and LA into the RA (Figure 12). Stage IIIB is closure of the fenestration (Figure 13G and Figure 14C), usually by transcatheter methodology with an atrial septal closing device [51,53,54,55] six to 12 months after Stage IIIA. Amplatzer Septal Occluder is most commonly used for this purpose [54,55].

### 10.4. Follow-Up after Palliative and Corrective Surgery

After palliation during the neonatal period and following multi-staged Fontan surgery, clinical and echocardiographic follow-up is necessary to detect interstage problems and post-Fontan complications, respectively. The inter-stage mortality rates varied from 5% to 15% [47,56,57]. These are more frequent between Stages I and II than between Stages II and III. The causes of interstage mortality have been identified and include obstructed atrial defect, narrowing of the aortic arch, stenosis or distortion of the pulmonary arteries, AV valve regurgitation, thrombosis or narrowing of the aorta-pulmonary shunts, and inter-current illnesses [56,58]. The care–giver should be cognizant of these issues and develop strategies to identify them by periodic clinical evaluation along with echo-Doppler and other imaging studies, as deemed appropriate and address them promptly. Such efforts are likely to prevent/lessen the morbidity and mortality. Discussion of diagnostic methods and treatment techniques to address these issues is beyond the scope of this paper and reader is referred elsewhere for the treatment of this subject [46,47,57,59].

Similarly, complications were observed during follow-up evaluation of patients who had all three stages of Fontan and these are arrhythmias; Fontan pathways obstruction; residual shunts; thrombo-embolic events, including, cerebro-vascular accidents or transient ischemic attacks; cyanosis; collateral vessels; and systemic venous congestion, including protein-losing enteropathy [44,45,60]. Luckily, such complications happen less often in subjects who were managed with the currently used staged TCPC with extra-cardiac conduit than children who had formerly used varieties [42,43] of Fontan surgery. Again, the care-giver should attempt to detect these complications and treat them as soon as they are identified. How these complications are identified and treated is reviewed elsewhere for the interested reader [44,46,60,61].

## 11. Summary and Conclusions

Mitral atresia with normal aortic root is a rare cardiac defect with hypoplastic LA and has a propensity to develop atrial septal restriction. The clinical presentation is largely dependent upon the associated heart defects. Echo–Doppler studies are useful in making the diagnosis and help guide the type of therapy. Babies with pulmonary atresia or severe pulmonary stenosis should undergo systemic to pulmonary arterial shunt, usually by a modified BT shunt. Early transcatheter or surgical atrial septostomy in patients with interatrial obstruction and placement of pulmonary artery band in patients with unrestricted pulmonary blood flow have become major components of early palliative treatment. As the infants grow, close attention should be paid to evaluate adequacy of atrial defect by serial echo studies. Atrial septostomy or septectomy may be needed later in infancy, if it was not required or not performed in the neonatal period. Then, staged Fontan, consisting of bidirectional Glenn and extra-cardiac conduit with fenestration with subsequent closure of fenestration should be undertaken, similar to those used in other single ventricle defects. Interstage and post-Fontan complications should be identified and promptly treated as appropriate.

## Figures and Tables

**Figure 1 children-09-01148-f001:**
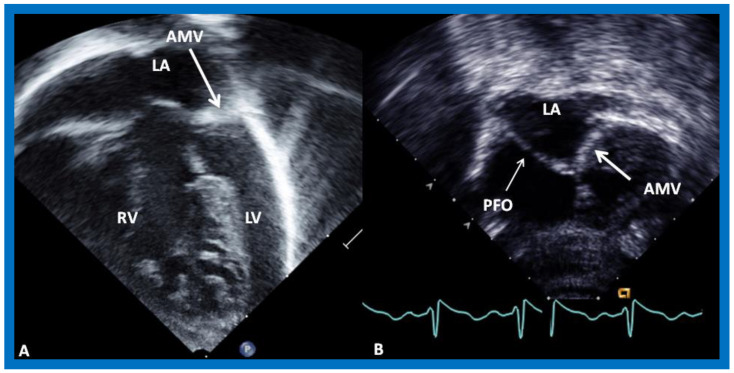
Selected video images from modified apical four-chamber views of two babies with mitral atresia illustrating atretic mitral valves (AMV), shown by thick arrows. The left atrium (LA) and left ventricle (LV) are small while and the right ventricle (RV) is large (**A**). A restrictive patent foramen ovale (PFO) is shown by a thin arrow in (**B**). Reproduced from [7].

**Figure 2 children-09-01148-f002:**
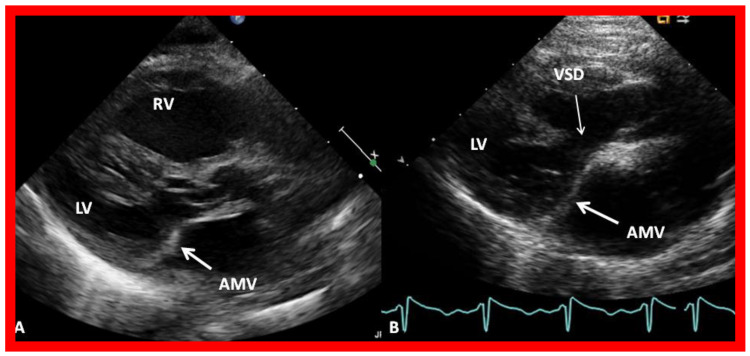
Selected video images from parasternal long axis views of two babies with mitral atresia illustrating atretic mitral valves (AMV), shown by thick arrows. A ventricular septal defect (VSD) is pointed out by a thin arrow in (**B**). The left ventricle (LV) is small while the right ventricle (RV) is large particularly demonstrated in (**A**). Reproduced from [7].

**Figure 3 children-09-01148-f003:**
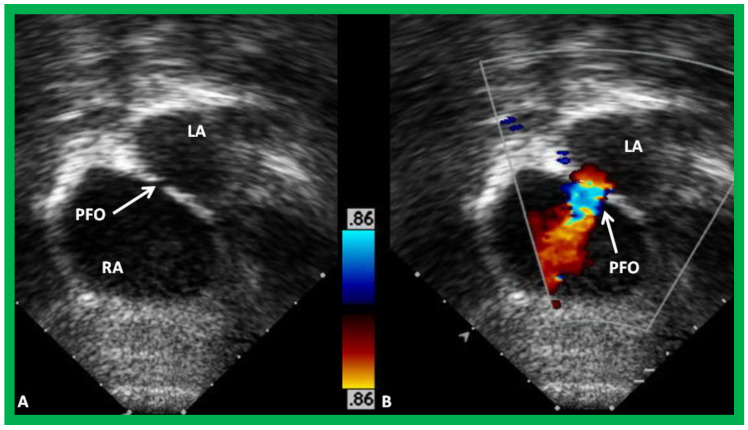
(**A**)**.** Selected video image from subcostal echocardiographic projection of a baby with mitral atresia demonstrating a small and restrictive patent foramen ovale (PFO) (arrow). (**B**)**.** Color flow Doppler illustrates color flow acceleration (arrow in (**B**)) across the PFO. LA, left atrium; RA, right atrium. Reproduced from [7].

**Figure 4 children-09-01148-f004:**
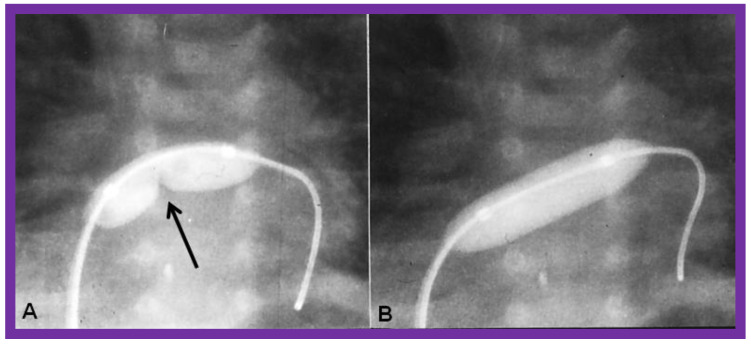
Selected cine frames of a static balloon angioplasty procedure to enlarge a restrictive patent foramen ovale, illustrating an inflated balloon in the posterior-anterior (**A**) view, displaying waisting of the balloon (arrow in (**A**)) which was fully abolished following further inflation of the balloon (**B**). Modified from [7].

**Figure 5 children-09-01148-f005:**
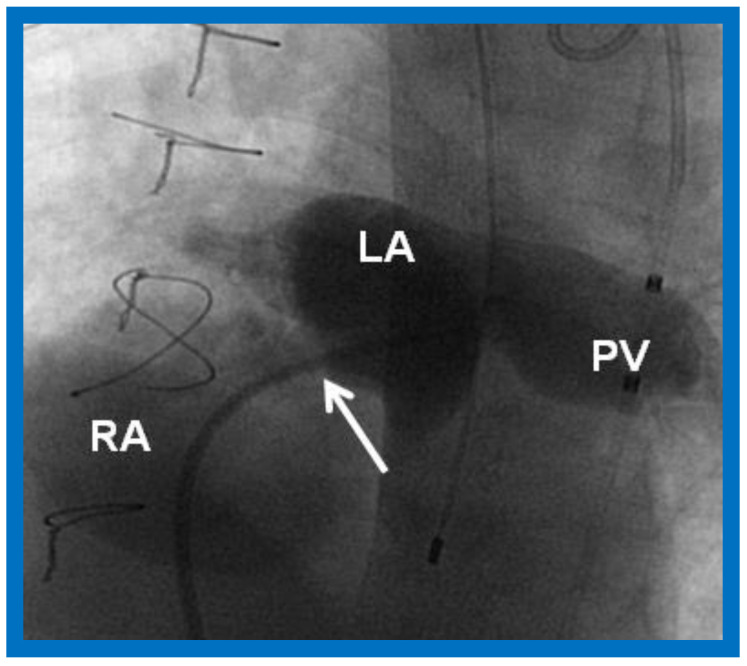
A selected cine frame from a cine-angiogram of the left atrium (LA) illustrates the position of the restricted atrial septum (arrow) and a markedly dilated pulmonary vein (PV). Sternal wires from prior surgery are seen, but not labeled. RA, right atrium. Modified from [7].

**Figure 6 children-09-01148-f006:**
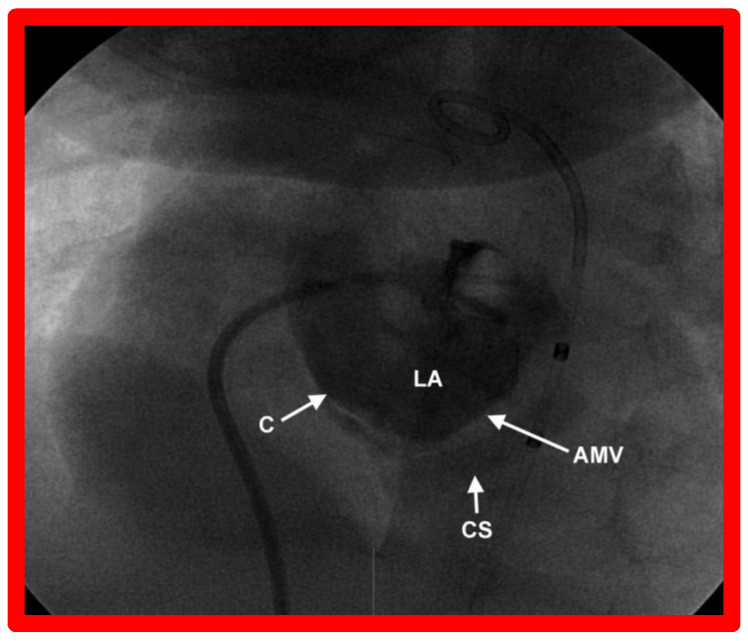
Selected cine image from a left atrial (LA) cine-angiogram in a left axial oblique (30° LAO and 30° cranial) view illustrating atretic mitral valve (AMV). Opacification of the coronary sinus (CS) is seen via a connecting (C) vein. Such communications, including levoatriocardinal veins [16] have been documented in the literature. Reproduced from [4].

**Figure 7 children-09-01148-f007:**
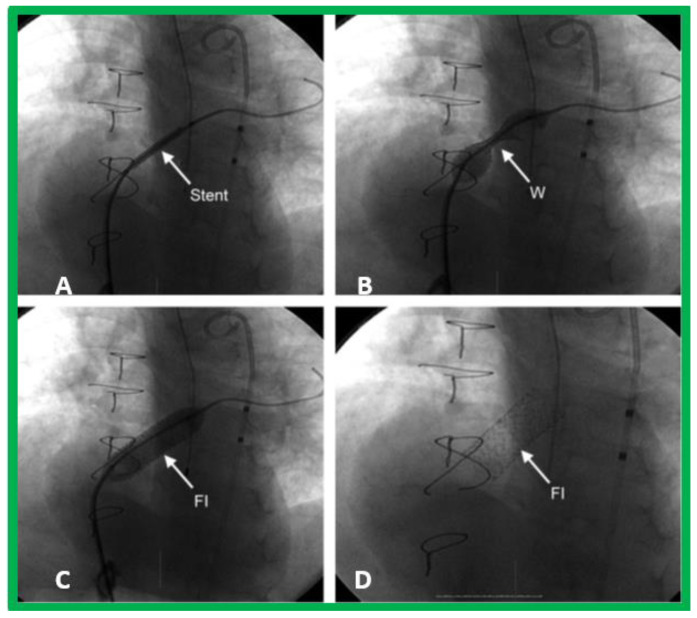
(**A**–**D**)**:** Selected cine frames while implanting the stent across a severely restrictive patent foramen ovale shown in Figure 5, illustrating the position of the stent prior to balloon inflation (**A**), during balloon inflation with a waist (W) of the balloon (**B**), and at the conclusion (**C**) of stent implantation. The position of the fully inflated stent (FI) before (**C**) and following (**D**) removal of the balloon is shown. Sternal wires from prior surgery are seen, but not labeled. PT, marker pigtail catheter. Modified from [4].

**Figure 8 children-09-01148-f008:**
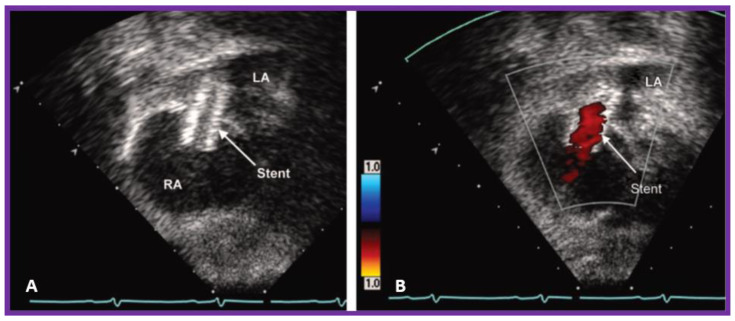
(**A**)**.** Selected 2D video image from subcostal view of an echo-Doppler study of an infant with mitral atresia who had a stent implanted 1 day prior to the study (Figure 7), illustrating the position of the stent (arrow) across the atrial septum. (**B**)**.** Same as (**A**) but with color flow mapping. Note laminar flow (arrow) within the stent suggesting non-obstructed flow across the atrial septum. LA, left atrium; RA, right atrium. Reproduced from [4].

**Figure 9 children-09-01148-f009:**
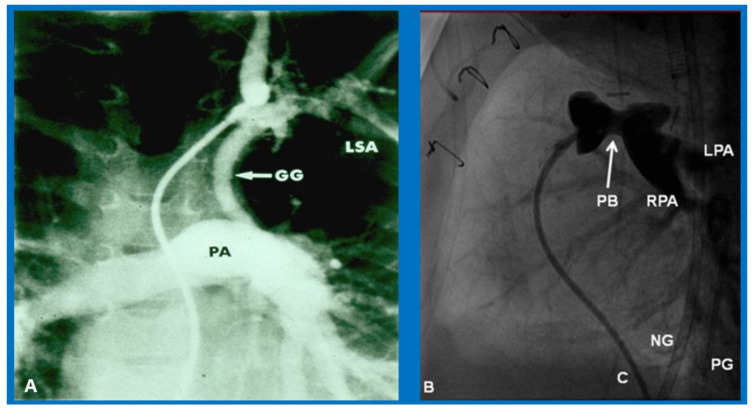
(**A**)**.** Selected cine-angiographic frame showing modified Blalock-Taussig (BT) shunt connecting left subclavian artery (LSA) with the pulmonary artery (PA) with a Gore-Tex grafts (GG) [30]. This cine frame shows wide-open BT shunt and excellent opacification of PA. (**B**). Selected cine frame form PA cine-angiogram in straight lateral projection demonstrating narrowed section of the PA produced by pulmonary artery band (PB) shown by arrow in (**B**) in an infant who had the PB [37]. C, catheter; LPA, left pulmonary artery; NG, nasogastric tube; PG, pigtail catheter; RPA, right pulmonary artery. Modified from [7].

**Figure 10 children-09-01148-f010:**
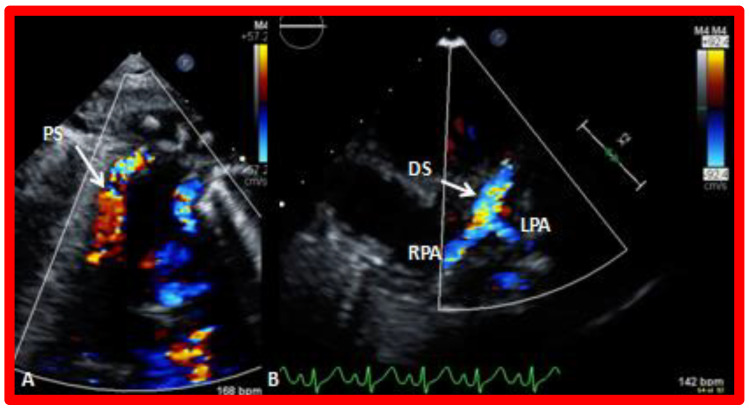
Selected video frames obtained from suprasternal notch view showing proximal shunt–(PS) by color flow imaging (**A**). In a slightly different view (**B**), the flow from the distal shunt (DS) into right (RPA) and left (LPA) pulmonary arteries is shown. Reproduced from [38].

**Figure 11 children-09-01148-f011:**
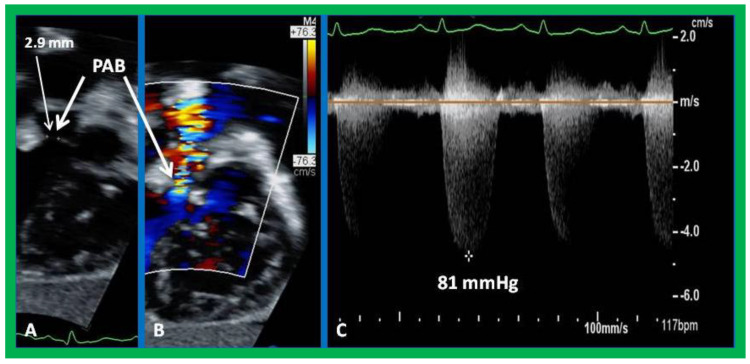
Selected echo–Doppler images demonstrating pulmonary artery band (PAB) which is narrow (2.9 mm) by two–dimensional imaging (**A**) and by color flow Doppler (**B**) and a substantial gradient (81 mmHg) by continuous wave Doppler interrogation (**C**). Reproduced from reference [38].

**Figure 12 children-09-01148-f012:**
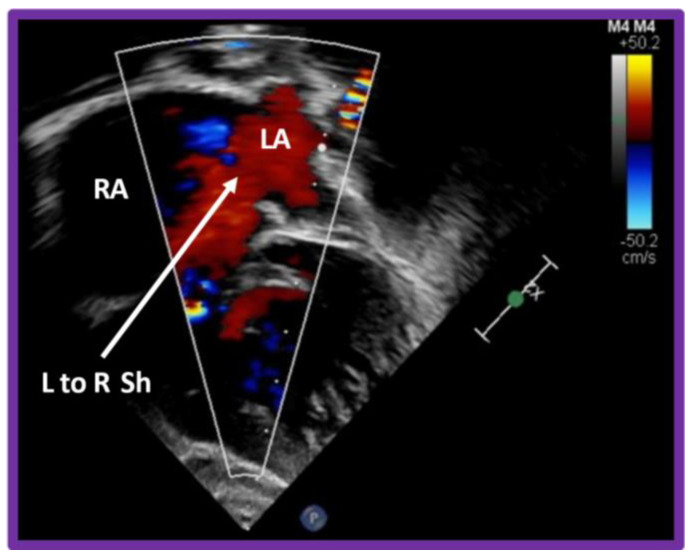
Selected video image from modified apical four-chamber views in a patient with mitral atresia illustrating unrestricted left to right shunt (L to R Sh) across the atrial setpal defect (arrow). Note the non–turbulent laminar flow. LA, left atrium; RA, right atrium.

**Figure 13 children-09-01148-f013:**
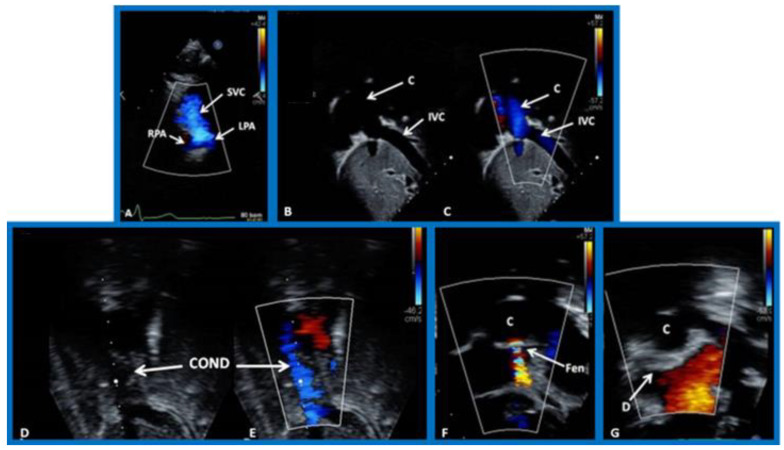
(**A**). Selected video images from suprasternal notch view demonstrating bidirectional Glenn shunt. The superior vena cava (SVC) is seen to empty into the right (RPA) and left (LPA) pulmonary arteries by color flow Doppler. (**B**,**C**). Selected video frames demonstrating connection between the inferior vena cava (IVC) and the conduit (C) by two-dimensional (**B**) and color flow Doppler (**C**). The IVC–C junction is widely patent. (**D**,**E**). Selected video frames demonstrating a patent conduit (COND) by two-dimensional (**D**) and color flow Doppler (**E**). Laminar flow as seen in (**E**) indicates that there no obstruction within the COND. (**F**). Selected video frames from an apical four chamber view by two-dimensional and color flow Doppler (**F**) illustrating a cross-sectional view of the C and fenestration (Fen). Turbulent flow is seen across Fen. (**G**). Selected video frame from apical four chamber view showing the position of the Amplatzer device (**D**) (arrow in (**G**)). No residual shunt is demonstrated. Modified from [38].

**Figure 14 children-09-01148-f014:**
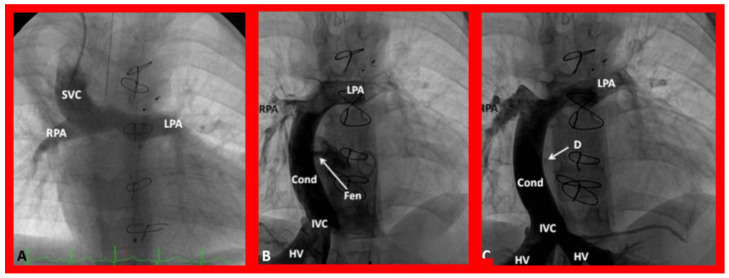
(**A**). Selected cine-angiographic frame showing bidirectional Glenn shunt (Stage II of Fontan). The unobstructed blood flow from the superior vena cava (SVC) into the right (RPA) and left (LPA) pulmonary arteries is seen. (**B**). Selected cine frame in antero-posterior view demonstrating Stage IIIA of the Fontan procedure conveying the inferior vena caval (IVC) blood flow into the RPA and LPA via a non-valve conduit (Cond). The fenestration (Fen) is shown by the arrow in (**B**,**C**). This cine frame demonstrates occlusion of Fen with an Amplatzer device (D), pointed out by the arrow in (**C**) (Stage IIIB). Sternal wires from prior surgery are seen in (**A**–**C**) and were not labeled. HV, hepatic veins. Modified from [44].

## Data Availability

Not applicable.

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
