# Peer review of "Mitral Atresia with Normal Aortic Root"

_children, 2022, doi:10.3390/children9081148_

Round 1
Reviewer 1 Report
1. A well-written manuscript
2. line 201: trial septum, should be atrial septum?
3. line 302: __ the Fontan: an extra space
Author Response
- A well-written manuscript – Thank you
- line 201: trial septum, should be atrial septum? – correct; is revised accordingly
- line 302: __ the Fontan: an extra space – revised accordingly
- The author thanks the reviewer for the diligent review and constructive suggestions.
Reviewer 2 Report
The paper gives an overall complete aproach to the diagnosis and management of this rare anomaly. I have only on point to change : In the paragraph Clinical presentation - page 2, Inter- atrial obstruction is indicated as 1.1 and the subsequent two subparagraphs Severe pulm. stenosis.. and No pulmonary stenosis are also indicated as 1.1.
I would suggest to change these two subparagraphs to 1.2 and 1.3. Otherwise I have nothing to object and I think the paper could be published after these minor corrections.
Author Response
The reviewer suggests: he paper gives an overall complete approach to the diagnosis and management of this rare anomaly. I have only on point to change: In the paragraph Clinical presentation - page 2, Inter- atrial obstruction is indicated as 1.1 and the subsequent two subparagraphs Severe pulm. stenosis. and No pulmonary stenosis are also indicated as 1.1. These are changed accordingly in the revised manuscript.
The author thanks the reviewer for the diligent review and constructive criticism.